# Acute Toxicity of Metal Oxide Nanoparticles—Role of Intracellular Localization In Vitro in Lung Epithelial Cells

**DOI:** 10.3390/ijms26178451

**Published:** 2025-08-30

**Authors:** Andrey Boyadzhiev, Sabina Halappanavar

**Affiliations:** 1Environmental Health Science and Research Bureau, Health Canada, Ottawa, ON K1A 0K9, Canada; andrey.boyadzhiev@hc-sc.gc.ca; 2Biology Department, Faculty of Science, University of Ottawa, Ottawa, ON K1N 9A4, Canada

**Keywords:** hyperspectral imaging, lysosomes, nanomaterials, co-localization, darkfield, fluorescence, nanoparticles

## Abstract

Endocytic uptake and lysosomal localization are suggested to be the key mechanisms underlying the toxicity of metal oxide nanoparticles (MONPs), with dissolution in the acidic milieu driving the response. In this study, we aimed to investigate if MONPs of varying solubility are similarly sequestered intracellularly, including in lysosomes and the role of the acidic lysosomal milieu on toxicity induced by copper oxide (CuO) nanoparticles (NPs), nickel oxide (NiO) NPs, aluminum oxide (Al_2_O_3_) NPs, and titanium dioxide (TiO_2_) NPs of varying solubility in FE1 lung epithelial cells. Mitsui-7 multi-walled carbon nanotubes (MWCNTs) served as contrasts against particles. Enhanced darkfield hyperspectral imaging (EDF-HSI) with fluorescence microscopy was used to determine their potential association with lysosomes. The v-ATPase inhibitor Bafilomycin A1 (BaFA1) was used to assess the role of lysosomal acidification on toxicity. The results showed co-localization of all MONPs with lysosomes, with insoluble TiO_2_ NPs showing the greatest co-localization. However, only acute toxicity induced by soluble CuO NPs was affected by the presence of BaFA1, showing a 14% improvement in relative survival. In addition, all MONPs were found to be associated with large actin aggregates; however, treatment with insoluble TiO_2_ NPs, but not soluble CuO NPs, impaired the organization of F-actin and α-tubulin. These results indicate that MONPs are sequestered similarly intracellularly; however, the nature or magnitude of their toxicity is not similarly impacted by it. Future studies involving a broader variety of NPs are needed to fully understand the role of differential sequestration of NPs on cellular toxicity.

## 1. Introduction

The engineered nanomaterials (ENMs; materials designed for specific applications exhibiting at least 1 dimension or structure in the size range of 1–100 nm) have become endemic in consumer markets over the last 20 years [1]. Engineered metal oxide nanoparticles (MONPs) represent one of the largest classes of ENMs used worldwide, which present a risk for occupational exposure through accidental inhalation when the material is manipulated in its dry, dusty state [2]. As with all ENMs, the toxicity of MONPs is influenced by particle-specific physicochemical properties, of which dissolution in biological environments is considered to be of importance [3,4,5].

In previous studies, we systematically investigated the role of material solubility in cellular toxicity induced by MONPs, including the toxicity of zinc oxide (ZnO), copper oxide (CuO), nickel oxide (NiO), aluminum oxide (Al_2_O_3_), and titanium dioxide (TiO_2_) nanoparticles (NPs) that exhibit varying solubility profiles and chemical composition [6,7]. The results showed that particulate and dissolved forms of all MONPs induced similar transcriptional responses in lung cells exposed under submerged conditions; however, the contribution of dissolution was dependent on the solubility of the material in the culture medium [7]. These studies also showed that extracellular dissolution contributed strongly to toxicity induced by highly soluble particles (showing >70% dissolution), such as ZnO. For soluble MONPs (showing 10–70% dissolution), such as CuO, both particle and ionic forms contributed to toxicity, while dissolution did not play a role in the toxicity of poorly soluble–negligibly soluble MONPs (showing ~1% dissolution) [7], such as TiO_2_.

CuO NPs, NiO NPs, and Al_2_O_3_ NPs have shown higher solubility in the caustic environment of lysosomes [8,9,10]. A study involving BEAS-2B bronchial epithelial cells noted that inhibition of lysosomal acidification using the v-type ATPase inhibitor Bafilomycin A1 (BafA1) significantly reduced the cytotoxic response induced by CuO NPs [11], suggesting that CuO NPs are taken up by cells prior to their dissolution in lysosomes, which is responsible for the toxicity observed. Other MONPs, such as iron oxide nanoparticles and labelled cerium oxide nanoparticles, have been directly observed localizing to lysosomes and endosomes in vitro [12,13]. For certain MONPs, such as ZnO and TiO_2_ NPs, particle sequestration in actin–lipid aggregates called ‘cauliflowers’ has recently been shown, which acts as both a nanoparticle reservoir as well as a mechanism for repeated exposure through a cycle of exocytosis and endocytosis [14]. While particle localization has been assessed independently for some MONPs in the past, it is not clear if intracellular particle localization is driven by their solubility profiles or if MONPs are transported and sequestered similarly in cells regardless of their solubility.

Determining the subcellular location of the ENM post-exposure can be a non-trivial endeavor, especially when dealing with unlabelled particles. Transmission electron microscopy (TEM) has been used because of its high sensitivity and resolution to detect particles in specific cellular compartments [15]. However, it is not conducive to routine analysis owing to the specialized skill set required for sample processing, microscope operation, and the costs involved. Enhanced darkfield microscopy with hyperspectral imaging (EDF-HSI) has been applied to locate unlabelled ENMs in both tissues in vivo [16,17,18] and cells in vitro [19,20,21]. When combined with fluorescence labelling, EDF-HSI enables simultaneous detection of the ENM and its cellular localization post-exposure. EDF-HSI with fluorescence has been utilized in the past to assess co-localization of silica (SiO_2_) NPs with Lysosomal-Associated Membrane Protein 1 (Lamp1), a lysosomal marker, in mouse lung epithelial cells [20], and more recently to show the uptake and localization of folate-conjugated SiO_2_ NPs in relation to the nucleus and actin cytoskeleton [22]. Furthermore, a combined EDF-HSI and 3D imaging approach has been applied to show direct interaction of multi-walled carbon nanotubes (MWCNTs) with lung cells in mice following in vivo exposure and in precision cut lung slices after ex vivo exposure [23,24].

The objectives of the present study were to (1) determine if CuO, Al_2_O_3_, NiO, and TiO_2_ MONPs that vary in their solubility are similarly sequestered in lysosomes and the relevance of vesicular acidification on their toxicity in lung cells in vitro and (2) utilize EDF-HSI in conjunction with fluorescence microscopy and 3D imaging to assess the degree of co-localization of the four individual MONPs with lysosomes and potentially with actin sequestering structures. In brief, lung epithelial cells (FE1) were exposed to individual MONPs with or without the lysosomal acidification inhibitor BafA1. Using immunocytochemistry, cellular sequestration and the potential impact on cellular structures were identified. Mitsui-7 (a type of MWNCT) was used as a non-metal control, which is known to induce cytoskeletal disturbances in vitro [25,26].

## 2. Results

### 2.1. Inhibition of Vesicular Acidification and Impacts on Cellular Viability

Treatment of FE1 cells with 1–200 nM BafA1 for 24 h induced a concentration-dependent reduction in viable cells, beginning at 5 nM (Figure 1a). LysoTracker Red DND-99 staining of untreated control cells showed strong, punctate, mainly perinuclear but non-uniform staining of lysosomes, the intensity of which significantly reduced in cells treated with different BafA1 concentrations, apparent at 10 and 25 nM, with diffuse cytoplasmic staining starting at 5 nM (Figure 1b). Due to the high toxicity of BafA1 in FE1 cells, a concentration of 5 nM was chosen for all subsequent exposures.

When FE1 cells were co-exposed for 24 h with 5 nM BafA1 and CuO, NiO, Al_2_O_3_, TiO_2_ NPs, or Mitsui-7 MWCNTs, an improvement in relative survival was only observed in cells exposed to 25 µg/mL CuO NPs (52% cell survival with 5 nM BafA1, 38% in the absence of BafA1) (Figure 2). Both conditions were statistically significantly different compared to dimethyl sulfoxide (DMSO)-treated cells but were not significantly different from each other. Cellular response to 54 µg/mL copper (II) chloride (CuCl_2_) did not change in the presence or absence of BafA1 (60–62% relative survival). For cells co-exposed to NiO, Al_2_O_3_, or TiO_2_ MONPs and 5 nM BafA1, an increase in cellular toxicity was observed in the presence of BafA1 compared to cells exposed to ENMs alone. With respect to NiO NPs, this difference was statistically significant (*p* < 0.01).

Since 5 nM BafA1 was able to induce a modest reversal of cellular toxicity and improve cell viability in cells exposed to CuO NPs, the effect of CuO NPs on acidic vesicle integrity in FE1 cells was tested using LysoTracker Red DND-99 staining (Figure 3). In control untreated cells, Lysotracker staining was pronounced and appeared as bright red foci (Figure 3). Noticeable reduction in LysoTracker Red staining was observed at 25 µg/mL CuO NPs, which co-located with the distribution of particles.

### 2.2. Hyperspectral Libraries for Particle Mapping

While nanoscale objects can be visualized to an extent using enhanced darkfield illumination, spectral matching is used to differentiate between non-specific bright objects from actual ENMs. Pure particle libraries for MONPs dispersed in ProLong mounting medium are shown in Appendix A. For CuO NPs, two separate spectral profiles were observed: one with a broad peak at 670 nm and the other at 720 nm. Similarly, for NiO NPs, two spectral profiles were observed, at 610 nm and at 645 nm. Al_2_O_3_ and TiO_2_ NPs were blue-shifted relative to CuO and NiO NPs, with a main broad peak at 570 nm for Al_2_O_3_ NPs and two spectral populations for TiO_2_ NPs at 540 nm and 580 nm.

### 2.3. Co-Localization of MONPs with Lamp1 and Impacts on α-Tubulin

Since treatment with BafA1 did not improve cytotoxicity induced by NiO, Al_2_O_3_, and TiO_2_ NPs, their association with lysosomes was assessed and compared with CuO NPs via Lamp1 staining.

Representative 40× images of Lamp1 and particle distributions in FE1 cells after 24 h exposure to 6 and 12.5 µg/mL of CuO, NiO, Al_2_O_3_, or TiO_2_ NPs are provided in Figure 4. The distribution of Lamp1 in control and all exposed cells appeared perinuclear, showing no change in distribution in response to MONP treatment at either concentration. MONPs can be identified as bright white objects in the images, which are absent in unexposed cells. Aggregates of MONPs can be seen associated with cells in all instances, with no obvious pattern in their organization.

To determine the degree of co-localization of MONPs and Lamp1 signals, the overlap between Lamp1 and darkfield channels in 100× oil immersion images was measured in ImageJ 1.54f. Co-labelling with α-tubulin was used to visualize the cellular boundaries. Figure 5 and Appendix A confirm that darkfield and Lamp1 signals showed subtle co-localization in all instances. For CuO and NiO NPs at 6 and 12.5 µg/mL and for 12.5 µg/mL Al_2_O_3_ NPs, only one image in three showed co-localization using this approach (Table 1). For CuO and Al_2_O_3_ NPs at 6 and 12.5 µg/mL and NiO NPs at 12.5 µg/mL, the maximum proportion of darkfield signal, which co-localizes with Lamp1 signal, is <10%. More pronounced co-localization was seen for TiO_2_ NPs, with 5/6 images showing signal co-localization. The maximum amount of co-localizing darkfield signal was concentration-dependent, with a maximum of 26.9% of the particle signal showing co-localization at 6 and 53.0% at 12 µg/mL. The identity of the co-localized bright objects was confirmed by hyperspectral imaging using pure particle libraries. Deconvolution of 100× 3D z-stacks captured for each image showed that the matched MONPs are in the same plane as the Lamp1 signal, and in close association (Appendix A).

### 2.4. Co-Localization of MONPs and F-Actin Aggregates

Labelling of FE1 cells with Phalloidin allowed for visualization of the F-actin cytoskeleton, which is implicated in endo/exocytosis activities and the formation of sequestering structures termed ‘cauliflowers’ in the pulmonary epithelium.

In all samples, including blank media treated, rounded aggregates of F-actin were observed (Figure 6). Cytoplasmic inclusions can be seen in phase contrast images, which appear at a similar size and density as fluorescently labelled F-actin aggregates, with a diameter of 2–4 µm (Appendix A). MONPs were always seen to associate with F-actin aggregates after MONP exposure. Particles showed an increased tendency to localize to these bright staining actin aggregates compared to the co-localization observed between Lamp1 and the particles. The 3D deconvolution of the areas surrounding the aggregates in Al_2_O_3_ NP and NiO NP samples showed that particles appear both in-plane and above the plane of the main F-actin cytoskeleton (Appendix A), which was also observed for CuO and TiO_2_ NPs.

### 2.5. High Concentrations of MONPs and MWCNTs Show Differing Effects on Actin and Tubulin Cytoskeleton, and Lamp1 Signal Distribution

To test for acute effects on F-actin and α-tubulin cytoskeleton structure, and on Lamp1 distribution following exposure to high concentrations of MONPs, cells were exposed to 25 µg/mL soluble CuO NPs and 50 µg/mL insoluble TiO_2_ NPs, with 100 µg/mL Mitsui-7 MWCNTs serving as an insoluble non-MONP and non-particle contrast.

Based on representative 100× images in Figure 7a, control cells exposed to blank media showed well-defined reticular tubulin meshwork inside the cell, with the main Lamp1 signal located in the perinuclear region. No change in Lamp1 distribution was seen in response to any treatments; however, differences were seen with respect to effects on the α-tubulin cytoskeleton. In response to 25 µg/mL CuO NPs, large holes were observed in the α-tubulin structure (Figure 7b). Conversely, exposure to 50 µg/mL TiO_2_ NP induced fragmentation of tubulin fibers as compared to blank media treatment (Figure 7c). In contrast, 100 µg/mL Mitsui-7 MWCNTs did not appear to induce differences in background tubulin structure (Figure 7d); however, in cells undergoing division, association of MWCNT fibers with microtubules was observed (Figure 7e). Deconvolution of 3D stacks of the fiber-laden dividing cell showed the presence of MWCNTs intersecting the tubulin fibers, surrounding the mitotic nuclei (Appendix A).

As compared to effects on α-tubulin structure, 24 h exposure to a high concentration of CuO NPs, TiO_2_ NPs, and Mitsui-7 MWCNTs also induced changes to the F-actin cytoskeleton (Figure 8). Cells exposed to blank media showed well-defined stress fibers distributed throughout the cell (Figure 8a). Treatment with 25 µg/mL CuO NPs induced the formation of holes in the actin structure, observed directly under particle aggregates (Figure 8b). Exposure to 50 µg/mL of insoluble TiO_2_ NPs resulted in similar stress fiber formation as in control cells, albeit with an increased incidence of areas featuring diffuse staining characteristic of unpolymerized actin (Figure 8c). Finally, with respect to 100 µg/mL Mitsui-7 MWCNTs, no impact on background F-actin structure was observed (Figure 8d); however, MWCNT-laden bi-nucleated cells lacking F-actin stress fibers were noted (Figure 8e).

Lastly, cells were exposed to 100 µg/mL Mitsui-7 MWCNTs with or without 5 nM BafA1 to determine if inhibition of lysosomal acidification influences the toxicity induced by high concentrations of Mitsui-7 MWCNTs, a high aspect ratio fiber, in a similar manner as that observed for CuO NPs. In Figure 9, it is shown that the reduction in cell viability in cells exposed to BafA1 alone or co-exposed to BafA1 and MWCNTs, in comparison to media controls, was the same (~80%).

## 3. Discussion

MONPs-induced toxicity is influenced partly by their solubility in biological environments, which in turn is influenced by particle size, chemical composition, and the pH of the biological medium. Previous studies have shown that for highly soluble ZnO, which dissolves instantaneously extracellularly, toxicity proceeds akin to a fully dissolved equivalent [27]. For MONPs that exhibit medium-low solubility, particle uptake and consequent intracellular dissolution in lysosomes are thought to play a key role in their toxic potential. Even for insoluble MONPs, such as TiO_2_, which are stable under acidic conditions [28], evidence in animal models exists for lysosomal localization and subsequent instability [29]. In the present study, intracellular localization, including localization to lysosomes, was investigated to understand if all MONPs are similarly sequestered and if cellular sequestration contributes to their toxicity.

### 3.1. Relevance of Lysosomal Dissolution on Toxicity

Based on a BafA1 dose range finding experiment (Figure 1a), FE1 cells were particularly sensitive to v-type ATPase inhibition. Studies using human lung epithelial cells have shown that in A549 and BEAS-2B cells, up to 24 h of exposure to concentrations as high as 100–200 nM BafA1 have no impact on cell viability [11,30]. At the low concentration of 5 nM BafA1 in FE1 cells, destabilization of lysosomes was already noticeable (Figure 1b), suggesting that FE1 cells are particularly sensitive to inhibition of lysosomal acidification. The high sensitivity of cells to BafA1 resulted in incomplete inhibition of lysosomal acidification, which may have contributed to the subtle changes in toxicity observed following co-exposures to BafA1 and particles.

From the four MONPs assessed (CuO, NiO, Al_2_O_3_, and TiO_2_), which vary in their intracellular and extracellular solubility, lysosomal dissolution over 24 h only contributed to the toxicity induced by CuO NPs (Figure 2 and Figure 3). A similar trend in toxicity, but with more pronounced protection against reductions in cell number, was observed following 24 h co-exposure with CuO NPs or CuCl_2_ and 100 nM BafA1 in BEAS-2B cells [11]. In another study, a concentration-dependent reduction in LysoTracker staining was reported in A549 lung cells exposed to CuO NPs for 3–6 h [31]. The modest improvement in cell viability in FE1 cells co-exposed to CuO NPs and BafA1 compared to other studies using human cells is due to the higher sensitivity of FE1 cells to BafA1 treatment, which limited the use of high BafA1 concentrations that are required to completely inhibit the acidification of lysosomes. The results of this study provide further weight of evidence in support of a role for lysosomal sequestration and consequent lysosomal dissolution of CuO NPs in the toxicity observed in lung cells.

Conversely, insoluble or poorly soluble TiO_2_, Al_2_O_3_, and NiO MONPs, and Mitsui-7 MWCNTs, showed an increased loss in cell viability when co-treated with BafA1, with indications of an additive response between the MONPs and the inhibitor (Figure 2). For TiO_2_ NPs, this response was expected, as they are shown to be insoluble and stable in lysosomal media [32]. With respect to Al_2_O_3_ NPs, while increased dissolution in acidic environments compared to neutral media has been reported [10], dissolved Al is unstable and forms stable precipitates in the presence of biomolecules found in cell culture medium [33]. NiO NPs have been shown to dissolve in cell culture medium and under acidic environments [9,34], with their dissolved form stable in the presence of bioligands. However, the results showed higher cytotoxicity in cells co-exposed to NiO NPs and BafA1. In addition to inhibiting lysosomal acidification, BafA1 is also known to block autophagic flux by inhibiting the fusion of autophagosomes and lysosomes [35]. At present, it is not possible to inhibit lysosomal acidification without also affecting autophagy in some capacity. With respect to NiO NPs, experiments in HeLa cells identified a protective role for autophagy in NiO NP-mediated cytotoxicity [36]. Thus, lack of protection against cellular toxicity in the absence of lysosomal dissolution may be due to decreased autophagic flux. Finally, for Mitsui-7 MWCNTs, BafA1 co-exposure resulted in the same level of toxicity as BafA1 alone, ~80% (Figure 2). MWCNTs in general are known to be stable in acidic conditions; however, they may leach metal impurities into the surrounding media [37]. Together, these results indicate lysosomal dissolution over a 24 h span does not contribute to the toxicity of Al_2_O_3_ NPs, nor does the acidic lysosomal lumen potentiate toxicity induced by TiO_2_ NPs, while a more complex interaction may be occurring with respect to poorly soluble NiO NPs.

### 3.2. Localization of MONPs to Lysosomes

ENMs in general are reported to be sequestered in lysosomes. To understand if differences in levels of MONP sequestration in lysosomes were responsible for differences observed in BafA1 co-exposures, EDF-HSI fluorescence microscopy was used to determine the degree of co-localization between MONPs and Lamp1, a lysosomal marker.

EDF-HSI analysis showed that all MONPs localize to lysosomes, with insoluble TiO_2_ NPs showing the greatest co-localization potential. There are a few possible explanations for this observation. When the field of view from all four MONPs is compared, it is clear that more of the field of view positively maps for TiO_2_ NPs as compared to the other three nanoparticles (Figure 5c and Figure 6c). It is possible that the scattering properties and consequently, the spectral profiles of CuO NPs, NiO NPs, and Al_2_O_3_ NPs, which have enhanced dissolution in acidic environments, and which also showed some level of dissolution in cell culture medium, are modified due to interactions in lysosomes or the cellular medium. A shift in HSI spectral profiles was reported for polymer-coated gold/copper sulfide particles suspended in media at different pH levels, relative to spectra in neutral pH media [38]. A similar high level of co-localization has been observed between poorly soluble SiO_2_ NPs and lysosomes in FE1 cells. SiO_2_ NPs are also known to be stable in lysosomes, as is seen for TiO_2_ NPs [32,39]. Another explanation is that EDF-HSI analysis is not sensitive enough to accurately detect all MONPs associated with lysosomes at the single-cell level. EDF-HSI has been validated with traditional techniques, such as Raman microscopy and electron microscopy, for its ability to detect MONPs in tissue [16], and it has been used in conjunction with automated detection algorithms for highly sensitive and specific detection of nano gold in mouse tissues following injection [40]. However, at a single-cell level, similar validation has not been carried out. Therefore, the co-localization seen should not be taken to represent the full breadth of association between Lamp1 and MONPs.

### 3.3. Localization of MONPs to F-Actin Aggregates

In order to understand if different ENMs are sequestered or trafficked differently, impacting their intracellular localization and dissolution, the F-actin cytoskeleton was stained with Phalloidin post-exposure to individual CuO, NiO, Al_2_O_3_, and TiO_2_ NPs [41].

Based on EDF-HSI fluorescence, summarized in Figure 6, and 3D deconvolution of Z-stacks in Appendix A, it can be seen that all MONPs traffic to and co-localize with bright actin aggregates that appear both within and above the focal plane of the cells (Figure 5). A mechanism for the formation of actin–lipid cauliflowers with similar characteristics has been described both in vivo in mice and in vitro in mouse LA-4 cells after exposure to MONPs [14]. In this study, endocytosed MONPs (ZnO, TiO_2_) interact with lipid molecules inside pulmonary epithelial cells to make an initial aggregate, which is then coated in actin and exocytosed out of the cell, decorating the plasma membrane in ENM–actin–lipid structures. In the present study involving FE1 cells, bright actin aggregates were seen in MONP-treated cells as well as in cells treated with blank media (Figure 6), suggesting that these structures may represent a physiological process for sequestering particles. FE1 lung epithelial cells have been described as having both type I and II alveolar epithelial characteristics, with distinct subcellular inclusions visible under phase contrast [42]. Based on the phenotype of FE1 cells, it is possible these aggregates represent a subtype of lamellar body or secretory vesicles, which may be produced in type II alveolar epithelial cells and are responsible for surfactant secretion in the lung. They are characterized by an acidic microenvironment with large concentrations of phospholipids [43]. Subtypes of actin-coated lamellar bodies have been characterized in rat lung cells; however, they are smaller in size and more abundant than the actin aggregates seen here [44]. Further work is necessary to fully characterize these structures in FE1 cells; however, based on the results shown here, these inclusions naturally present in the cytoplasm may act to sequester particles upon their uptake in addition to lysosomes.

### 3.4. Differential Effects of Soluble CuO and Insoluble TiO_2_ NPs on Lysosomes and Cytoskeleton Structure

In order to understand the effects of cellular sequestration of particles on cytoskeletal structures and lysosome dynamics, FE1 cells were exposed to high concentrations of soluble CuO NPs or insoluble TiO_2_ NPs, shown to induce a cytotoxic response in vitro [6,7]. Mitsui-7 MWCNTs were included as a non-MONP and non-particle contrast (Figure 7 and Figure 8).

Changes in Lamp1 abundance or distribution were not seen in cells treated with any ENM (Figure 7). While 20 µg/mL CuO NPs have been shown to result in increased Lamp1 staining in human endothelial cells following 24 h of exposure [45], it is possible that the response takes longer to manifest in FE1 cells. With respect to TiO_2_ NPs, studies have reported an increase in lysosomal membrane permeabilization after 72 h of incubation and that long-term exposure to TiO_2_ NPs leads to a blockage in autophagic flux [46]. While clear evidence for changes to lysosome dynamics via Lamp1 staining was not observed in the present study, transcriptional responses in FE1 cells post-exposure to CuO or TiO_2_ NPs showed enrichment and activation of the ‘Autophagy’ and ‘CLEAR Signaling Pathway’ canonical pathways at 24 h, at the same concentrations tested here ([7], Supplementary Files 2 and 5), with CuO NPs inducing the buildup of vesicles within the cytoplasm after 48 h of exposure [6]. Together, these responses indicate that transcriptomic changes in response to MONPs are a sensitive indicator of lysosomal disturbances, which manifest prior to apical changes in Lamp1 abundance or distribution, and that the post-exposure sampling time used in the study may need further optimization for capturing the manifested events. Similar associations can also be made between the differential effects seen on the cytoskeleton.

Both CuO NPs and TiO_2_ NPs were also observed to induce disturbances in F-actin and α-tubulin organization at high concentrations (Figure 8). The large holes in the tubulin cytoskeleton in cells exposed to CuO NPs may indicate an indirect result of its toxicity, whereas the holes in the F-actin cytoskeleton could simply be due to optical occlusion from a dense particle cluster directly over top of the actin cytoskeleton, as the hole was the same size as the particle cluster. Similar damage to the tubulin cytoskeleton of FE1 cells was seen after 24 h exposure to 12.5 µg/mL silica nanoparticles [20]. At the concentrations tested in this study, TiO_2_ NPs and not CuO NPs, enriched multiple actin-related pathways at 48 h ([7], Supplementary Files 2 and 5). These were the ‘Regulation of Actin-based Motility by Rho’, ‘Actin Cytoskeleton Signaling’, and ‘Actin Nucleation by ARP-WASP Complex’ canonical pathways, which were enriched at 25, 50, and 100 µg/mL with a concentration-dependent increase in the number of differentially expressed genes in each pathway. No tubulin-related pathways were available in the ingenuity pathway analysis database as of 2023, precluding this specific comparison. Disorganization of actin was associated with the endocytic uptake of TiO_2_ NPs in H9c2 cardiomyoblasts [47]. Furthermore, disorganization and disruption of both actin and tubulin networks have been seen in SaOS-2 osteoblast-like cells [48], and more recently in A549 lung epithelial cells [49] exposed for 24 h to TiO_2_ NPs. While the mechanism behind this interference is not fully understood, evidence suggests that TiO_2_ NPs can directly bind to microtubules and induce conformational changes that decrease tubulin polymerization [50]. It is possible that a similar physical mechanism may be at play with respect to F-actin destabilization. Overall, the results of this study indicate that insoluble TiO_2_ NPs, but not soluble CuO NPs, can impair the stability or polymerization of both actin and tubulin.

In comparison to MONPs, Mitsui-7 MWCNTs did not induce noticeable changes in Lamp1 mobilization or F-actin and α-tubulin cytoskeletal structure (Figure 7d and Figure 8d). However, Mitsui-7 fibers were observed interacting with the mitotic spindle in dividing cells, and fiber-laden bi-nucleated cells were observed lacking actin stress fibers (Figure 7e and Figure 8e). Transcriptional response characterization in FE1 cells exposed to 100 µg/mL Mitsui-7 MWCNTs showed that the GO biological process ‘extra cellular matrix organization [GO: 30198]’ and ‘cell cycle arrest [GO: 7050]’ were uniquely enriched at this concentration, along with ‘regulation of cell proliferation [GO: 42127]’ [51]. Exposure to Mitsui-7 MWCNTs induced aneuploidy and cell cycle arrest in BEAS-2B lung epithelial cells, with MWCNTs surrounding and even penetrating the cell nuclei [52]. These results indicate that cellular interaction with MONPs vs. MWCNTs is different and that the ENM aspect ratio and size influence their toxicity in addition to their solubility.

### 3.5. Limitations and Future Directions

It is important to note a few limitations of the present study that will help refine future study designs. First, the study investigated a relatively limited set of MONPs. While the results provide indications that insoluble MONPs present enhanced lysosomal localization compared to MONPs that exhibit medium or low levels of solubility, this needs to be confirmed with a larger repertoire of particles, particularly insoluble MONPs of different chemical composition.

Second, there is an inability to completely inhibit vesicular acidification due to the high sensitivity of FE1 cells to BafA1. Enhanced sensitivity to BafA1 has been reported in certain leukemia cell lines and attributed to inhibition of both the early and late stages of autophagy, with concurrent activation of apoptosis [53]. Interestingly, the same concentrations shown to be toxic in these leukemic cell lines were non-toxic in normal hematopoietic cells. A similar effect could be occurring in the FE1 cell line, which are normal non-cancer epithelial cells spontaneously immortalized following repeated subculture [42]. Since inhibition of v-ATPase activity can increase cellular pH, it is also possible that FE1 cells are more sensitive to the effects of acidosis, potentially due to a compromised or reduced ability to maintain pH homeostasis. Optimizing experimental conditions to attain a better inhibition of ATPase activity in cells would allow for a clear understanding of the role of lysosomal sequestration and acidification in the toxicity of ENMs.

Finally, this study showed that ENMs may be sequestered in other organelles or cellular structures, such as actin structures. However, this study was not able to conclusively determine the composition of the structures where ENMs were localized. Detailed characterization of these structures would improve our understanding of particle trafficking and sequestration mechanisms that may be common between cell types.

Our previous investigations have identified differential impacts of solubility on the toxicity of MONPs and shown that extracellular solubility is not sufficient to explain their toxicity [6,7,54]. In this study, we investigated the relevance of lysosomal acidification and cellular localization on acute toxicity induced by CuO, Al_2_O_3_, NiO, and TiO_2_ NPs of varying solubility. All MONPs were found to co-localize with the Lamp1 lysosomal marker, with insoluble TiO_2_ NPs, which are stable in neutral and acidic environments, showing the greatest co-localization potential. However, only soluble CuO NPs showed an improvement in viability after inhibition of lysosomal acidification, implying that for less soluble MONPs, such as NiO and Al_2_O_3_ NPs, intracellular dissolution does not contribute to cytotoxicity for up to 24 h of exposure. In addition, all MONPs were observed to localize to large F-actin aggregates within cells following exposure. At 24 h, 50 µg/mL insoluble TiO_2_ NPs impacted the stability of F-actin and α-tubulin cytoskeletons. Mitsui-7 MWCNT, which were included as a non-metal oxide and a non-particle ENM, showed no effects on cytoskeleton structure but were notably present inside bi-nucleated cells and associated with the mitotic spindle in dividing cells. Thus, the results suggest that cellular uptake, localization, and intracellular dissolution impact ENM’s potential to induce toxicity but to different extents and via dissimilar mechanisms.

## 4. Methods

### 4.1. Materials and Reagents

Four individual MONPs, CuO, NiO, Al_2_O_3_, and TiO_2_ NPs, as well as Mitsui-7 MWCNTs, were included in the study, and their physicochemical properties are listed in Table 2. For NiO, Al_2_O_3_, and TiO_2_ NPs, the primary particle sizes were very similar, with mean sizes of 23.9–27.3 nm, whereas CuO NPs were larger, with a mean size of 64.8 nm. The aspect ratios indicate that CuO, NiO, and TiO_2_ NPs are roughly spherical, whereas the Al_2_O_3_ NPs have a rod-like morphology. Similarly, CuO, NiO, and TiO_2_ NPs have specific surface areas (SSAs) in the range of 10.3–52.7 m^2^/g, whereas Al_2_O_3_ NPs have an SSA of 145.3 m^2^/g. Based on 24 h dissolution experiments conducted in the cell culture medium used for toxicity testing at a 10 µg/mL concentration following the Organization for Economic Cooperation and Development (OECD) guidelines on dissolution testing of ENMs [55], CuO NPs are considered soluble (10–70% dissolution), Al_2_O_3_ NPs as poorly soluble (1–10% dissolution), and NiO and TiO_2_ NPs as negligibly soluble (<1% dissolution). The characteristics of Mitsui-7 MWCNTs used in this study have been reported in previous publications, revealing a range of sizes (Table 2). Based on a summary provided in [24], Mitsui-7 MWCNTs have tube diameters in the range of 40–100 nm, whereas the lengths range from 3 to 9.4 μm, with the aspect ratio between 30 and 235 and a specific surface area between 22 and 28 m^2^/g. No additional characterization work was carried out on these materials in the present manuscript. Transmission electron microscopy images of all MONPs used in this study [6,7,56] and scanning electron micrographs of the Mitsui-7 MWCNTs [51] have been previously published.

A list of all reagents used in this study, along with manufacturer and catalogue number, can be found in Table 3. Lyophilized Bafilomycin A1 was dissolved in 100% DMSO to a stock concentration of 32 µM. Stock vials were aliquoted and stored at −20 °C in the dark away from light. CuCl_2_ was dissolved in DNAse/RNAse-free ultrapure water at a 5 mg/mL stock concentration, followed by sterile filtration and storage at 4 °C.

### 4.2. Cell Culture and Maintenance

Immortalized mouse lung cells (FE1), with characteristics of type I and II alveolar epithelial cells, were used. FE1 cells have been utilized to conduct high-content global gene expression analysis [6,7] and genotoxicity screening [54,56] of the MONPs tested in the present study, as well as in cellular uptake and localization studies involving SiO_2_ NPs [20]. These cells have also been utilized by other groups to screen for genotoxicity induced by ENMs in vitro [58,59,60,61].

FE1 cells were maintained in Dulbecco’s Modified Eagle’s Medium Nutrient Mixture: F12 HAM (1:1) phenol red-containing medium and supplemented with 2% fetal bovine serum (FBS), 100 U/mL penicillin G & 100 µg/mL streptomycin, and 1 ng/mL human epidermal growth factor. Cells were cultured in a 37 °C incubator with 95% humidity and 5% CO_2_.

### 4.3. ENM Suspension

CuO, NiO, and Al_2_O_3_ NPs were prepared for exposure as described in [56], whereas TiO_2_ NPs were prepared according to [7]. Methods described in [24] were followed to prepare Mitsui-7 MWCNTs. All materials were weighed out on an XSR105 analytical mass balance (Mettler Toledo, Mississauga, ON, Canada) and suspended in dH_2_O using a Branson Ultrasonics Sonifier™ 450 (Crystal Electronics Inc., Newmarket, ON, Canada) on an ice bath according to the sonication conditions listed in Table 4. For all materials, a ½ inch disruptor horn and a removable flat tip were used, with the tip of the probe immersed 1.5 inches deep into the solution. For TiO_2_ NPs, a removable extension was used for sonication to allow the flat tip to reach the appropriate depth. For Mitsui-7 MWCNTs, the fibers were suspended in ultrapure water +2% FBS and shaken vigorously by hand for 15 min to allow for fibers to enter suspension prior to initial sonication. The suspended Mitsui-7 stock was frozen at −20 °C until use. The stock was thawed just before cell exposure and mixed by gentle pipetting prior to resonication and dilution.

### 4.4. Cell Seeding

For all exposures carried out in this study, FE1 cells were seeded at a density of 125–130,000 cells/well in 6-well plates. For Trypan Blue exclusion analysis, cells were seeded directly onto the well plates. For microscopy analyses, cells were seeded on 18 mm × 18 mm #1 thickness sterile glass coverslips in each well of a 6-well plate. In each case, sonicated ENMs in ultrapure water were diluted into fully supplemented DMEM/F12 without phenol red and used for 24 h exposures. Cells were allowed to recover for 24 h after initial seeding before treatment with ENMs.

### 4.5. Trypan Blue Exclusion Assay

Following 24 h of exposure, cells were washed once with 0.5 mL phosphate-buffered saline (PBS), dissociated using 150 µL Trypsin (3 min incubation, 37 °C, 95% humidity, 5% CO_2_), and resuspended in 0.5–1 mL of fresh phenol red-free cell culture medium (depending on cell density as assessed via phase contrast at the time of harvest). A total of 10 µL of cell suspension from control or exposed samples was combined with 10 µL of 0.4% Trypan Blue dye and incubated at room temperature for 5–10 min before manual cell counting on a hemocytometer. The relative survival was calculated using Equation (1).Relative survival = (exposed viable cell count [white]/control viable cell count [white]) × 100(1)

### 4.6. LysoTracker Red DND-99 Staining

To assess the integrity of acidic vesicles, LysoTracker Red DND-99 staining was used. After 24 h of exposure, the media was removed and FE1 cells were washed with 1 mL fresh, prewarmed phenol red-free media. Next, fresh phenol red-free media complemented with 60 nM LysoTracker Red DND-99 was added to each well, and the plates were incubated under light occlusion for 1 h. The plates were washed with 1 mL of fresh media and with 1 mL of PBS. Cells were fixed with 4% formalin in PBS for 11 min, washed 3 times with fresh PBS, and mounted onto slides for analysis using ProLong™ Glass Antifade Mountant with NucBlue™ counterstain. Slides were allowed to cure for 24 h at room temperature in the dark before storage in the dark at 4 °C for analysis.

### 4.7. Bafilomycin A1 Dose Range Finding

A dose range finding study was carried out to identify the appropriate concentration of BafA1 to achieve the desired inhibition of lysosomal acidification in FE1 cells without causing overt toxicity. Cells were seeded for Trypan Blue exclusion staining, as described in Section 4.4, and were exposed to 1.8 mL of 1–200 nM BafA1 in fully supplemented phenol red-free cell culture medium for 24 h. Cells exposed to medium only or medium containing 0.625% DMSO were used as negative controls. Trypan Blue analysis was carried out as described in Section 4.5, with one biological replicate assessed for each concentration in technical duplicates.

In parallel, cells were seeded for imaging analysis as described in Section 4.4, and acidic vesicles were labelled as described in Section 4.6 with LysoTracker Red following 24 h exposure to 1–25 nM BafA1. Cells treated with 0.02% DMSO served as a solvent control.

### 4.8. Bafilomycin A1 Nanomaterial Co-Exposures and Viability Analysis

To determine the impact of inhibition of vesicular acidification on acute MONP cellular toxicity, FE1 cells were seeded for Trypan Blue exclusion analysis as described in Section 4.4 and co-exposed to 25 µg/mL CuO NPs, 50 µg/mL NiO NPs, 50 µg/mL Al_2_O_3_ NPs, or 100 µg/mL TiO_2_ NPs with or without 5 nM BafA1 for 24 h. The concentrations of MONPs chosen for exposure have been shown to result in reductions in relative survival at 24 h in FE1 cells [7]. A concentration of 54 µg/mL CuCl_2_ (equivalent to dissolved copper concentration from 25 µg/mL CuO NPs) was utilized to represent fully dissolved copper metal. Furthermore, a concentration of 100 µg/mL Mitsui-7 MWCNTs was utilized as a non-metal oxide non-particle contrast for comparison. This concentration induced a pronounced transcriptional response in FE1 cells and represents the highest concentration tested in vitro [51]. In each experiment, a blank media control and a DMSO solvent control were included. All particles were sonicated according to Section 4.3 (Table 4).

Following exposure, Trypan Blue exclusion staining was conducted according to Section 4.5. For each condition, 2 technical and 3 biological replicates were conducted. Significant differences in relative survival (using the density of viable cells measured as cells/cm^2^) were compared in R, with the ‘multcomp’ package, using a one-way ANOVA with Tukey’s post hoc in the case of significant results. Normality and variance structure were confirmed for each sample set using the Shapiro–Wilk and Levene tests, respectively, prior to ANOVA computation.

### 4.9. Measuring CuO NP-Induced Vesicular Instability

CuO NP particle suspension and cell seeding were conducted as described in Section 4.3 and Section 4.4. Cells were treated for 24 h with blank media or 6, 12.5, and 25 µg/mL CuO NPs. Lysotracker Red staining was conducted according to Section 4.6.

### 4.10. Lysosome and Cytoskeleton Labelling

To test for particle localization to lysosomes and the effects of particle exposure on the cytoskeleton, FE1 cells were seeded for imaging analysis (Section 4.4). Following overnight recovery, cells were exposed to 6 or 12.5 µg/mL CuO, NiO, Al_2_O_3_, and TiO_2_ NPs. A concentration of 12.5 µg/mL had been used in the past to conduct co-localization analysis of SiO_2_ NPs with Lamp1 in FE1 cells [20], while a lower concentration of 6 µg/mL was included in the case that 12.5 µg/mL presented too many particles for analysis.

In a parallel experiment, FE1 cells were seeded for imaging analysis as in Section 4.4. After overnight recovery, cells were exposed for 24 h to 100 µg/mL Mitsui-7 MWCNTs, 50 µg/mL TiO_2_ NPs, or 25 µg/mL CuO NPs. Mitsui-7 MWCNTs were included as a non-MONP contrast that has been shown to induce cytoskeletal instabilities [25,26]. The concentrations of MONPs chosen are known to induce cytotoxicity or strong transcriptional signalling in FE1 cells, with between 10 and 77% relative survival at 24 h [7,51]. Cells were used for immunofluorescence experiments with Lamp1, F-actin, and α-tubulin antibodies according to Section 4.11 and imaged as described in Section 4.12, Section 4.13 and Section 4.14.

### 4.11. Immunofluorescent Staining with Lamp1, F-Actin, or α-Tubulin

Following 24 h FE1 cell exposure, glass coverslips were washed 2 times with fresh, warm media followed by 1 wash with PBS. Cells were fixed with 4% formalin in PBS for 11 min. Cells were then washed 3 times with fresh PBS, followed by permeabilization using 1% Triton X-100 in PBS for 5 min before antibody labelling. In all cases, fixation, permeabilization, and staining were conducted at room temperature (21 °C).

Lamp1 + α-tubulin staining. Cells were blocked with 0.1% Tween-PBS (PBST) solution containing 1% BSA for 30 min. Cells were then labelled with Anti-Lamp1 antibody at a 1/1000 dilution in 1% BSA + PBST solution for 1 h. Cells were washed 3 times with fresh PBS and then incubated with Goat Anti-Rabbit IgG H&L at a 1:200 dilution in PBST BSA solution for 1 h in the dark. Cells were washed 3 times with fresh PBS, transferred to a new plate, and blocking was repeated one more time for 30 min in PBST BSA solution. Cells were next incubated with anti-alpha tubulin antibody (Rabbit monoclonal) at a 1:1000 dilution in PBST BSA solution for 1 h in the dark. Cells were washed 3 times with fresh PBS and then labelled with Goat Anti-Rabbit IgG H&L at a 1:200 dilution for 1 h in the dark. Cells were washed three more times with PBS and mounted onto slides using hard-curing ProLong™ Glass Antifade Mountant with NucBlue™ Stain (containing Hoechst 33342 nuclear counterstain). After 24 h of curing at room temperature in the dark, slides were placed at 4 °C until analysis.

F-actin staining. Cells were blocked using PBST BSA solution for 30 min, after which cells were incubated with Phalloidin-iFluor 594 Reagent at a 1:1000 dilution in PBST BSA. Incubation time was optimized, and 30 min to 1 h was used for incubation with the primary antibody. Coverslips were washed 3 times with fresh PBS and were mounted onto slides using ProLong™ Glass Antifade Mountant with NucBlue™ Stain (containing Hoechst 33342 nuclear counterstain). After 24 h of curing at room temperature in the dark, slides were placed at 4 °C until analysis and kept away from light.

### 4.12. Enhanced Darkfield Hyperspectral Imaging

All imaging was conducted on a Cytoviva 3D enhanced darkfield hyperspectral microscope (Cytoviva Inc., Auburn, AL, USA), with an attached dual-mode fluorescence module with DAPI, FITC, and Texas Red filters installed. All darkfield and fluorescence images were captured on a monochrome QI825 camera (Teledyne Vision Solutions, Acton, MA, USA) to allow for channel merging. Fluorescence and 40× darkfield signals were captured using an Xcite Series 120Q light source (Excelitas Technologies Corp., Montreal, QC, Canada). Hyperspectral images (HSI, HSIs for plural) and darkfield images at 60 and 100× were captured using a Fiber-Lite DC-950 light source (Dolan-Jenner Industries, Boxborough, MA, USA).

Particle spectral library preparation. To create pure particle libraries for mapping purposes, aliquots of suspended and sonicated particles were diluted in ProLong mounting medium, deposited onto glass slides, and coverslipped. The slides were allowed to cure for 24 h at room temperature in the dark, after which at least 3 hyperspectral images per slide were captured using a 100× oil immersion objective. For each particle, the darkfield light source was set to 50% total illumination, the collar on the objective was reduced to its smallest diameter (NA = 0.6), and the exposure conditions were set to 0.25 s exposure, with 501 lines per image. In order to remove non-specific spectra from the libraries and control for artifacts, images of blank dH_2_O diluted in ProLong medium were also captured with the same conditions used for particle imaging. The initial pure particle libraries created using the ‘Particle Finder’ algorithm were filtered at a spectral angle mapping angle of 0.1 radians using control HSIs of dH_2_O in ProLong.

Imaging sequence. The images were taken in the following sequence. First, 40× images were taken using an air objective of the whole coverslip across 9 fields of view covering the full surface of the slip. For each field of view, the fluorescence images were taken in DAPI (nuclear), FITC (Lamp1), and Texas Red (α-tubulin or F-actin) channels. Next, the filter wheel on the Dual Mode Fluorescence module of the Cytoviva was moved between filter positions, bathing the sample in white light and producing darkfield illumination. Due to the scattering properties of MONPs and MWCNTs, they can be clearly seen against the dark background. A short exposure image (5 ms) was taken to serve as the darkfield channel. This was repeated for 9 separate fields of view. The darkfield condenser was adjusted at each new field of view in order to maintain maximal fluorescence and scattering intensity.

After 40× imaging, higher magnification images were captured at 60 and 100× using oil immersion objectives, in which 3 fields of view per coverslip were analyzed. At each field of view, the 60× image was taken, with the fluorescence channels imaged first (collar fully open, NA = 1.25), followed by the darkfield channel imaging with the collar fully closed (NA = 0.65). Once the 60× image was collected, the 100× objective was placed, and the imaging was repeated for the 60× objective, with NA = 1.3 for fluorescence imaging and 0.6 for darkfield imaging. The darkfield light source was set to 50% or 35% power at 60 and 100× (depending on the strength of scattering of the ENM); however, for imaging control cells, the light source was always set to 50% intensity, as this produces the stronger signal. At 60×, 5 ms of exposure time was used, whereas at 100×, 20 ms of exposure time was used. During the 100× imaging, HSI imaging proceeded darkfield image acquisition. For HSI acquisition, the darkfield light source was set to 50% intensity in all cases, the collar was fully closed on the 100× objective (NA = 0.6), the exposure time was set to 0.25 s, and the line number was set to 501. For identification purposes, HSIs from cells exposed to the material of interest were mapped using the SAM algorithm and filtered spectral libraries using a critical angle of 0.1 (same as what was used to filter the reference libraries). The final 2D fluorescence images were created by combining the fluorescence and darkfield channels in ImageJ 1.54f; however, the HSI imaging could not be directly overlaid due to small differences in the field of view, and it is shown side by side for comparison instead.

### 4.13. Two-dimensional Particle/Lamp1 Co-localization

To determine in an objective manner the degree of overlap between the signals for particles and Lamp1, the ‘Colocalization Threshold’ plugin in Fiji was used. This plugin utilizes an objective statistical approach originally published in [62] to conduct thresholding on two separate fluorescent channels and then determine the degree of pixel overlap from both channels. Statistical significance of this overlap was not determined. For Lamp1 images, a rolling ball background subtraction was conducted (50-pixel ball width) prior to co-localization analysis. Three separate 100× fields of view were analyzed per sample. Pixels of overlap between the two channels are presented in white in the generated maps.

### 4.14. Three-dimensional Imaging

At each high magnification field of view, 3D image stacks of fluorescence and darkfield channels were captured using a 100× oil immersion objective, with the collar fully open for fluorescent images (NA = 1.3) and fully closed for darkfield images (NA = 0.6). Z-stack size and imaging parameters varied depending on the specific field of view imaged. For MONP localization, ‘Just Locate Nanoparticles’ was used, which determines the location of spherical particles based on the slice showing maximal intensity for areas with scattering intensity past a certain threshold. NPs are visualized as red, 4-pixel spherical voxels in the 3D space. Fluorescence deconvolution was conducted in ImageJ using Cytoviva’s 3D analysis plugins. For deconvolution, point spread functions (PSFs) were generated based on the emission wavelength of the fluorophore in question for each target (Lamp1: 520 nm; α-tubulin: 603 nm; Hoechst 33342: 454 nm; Phalloidin: 618 nm). To create the final 3D visualizations, the NP localization file and the 3 deconvoluted fluorescence channels were opened in the ‘3D Viewer’ plugin in ImageJ and overlayed. In the case of Mistui-7 MWCNTs, a PSF was generated based on the main peak of the lamp spectrum of the darkfield light source used for channel imaging (661 nm). This PSF was used to deconvolute the channel in the same manner as for the fluorescent channels.

## Figures and Tables

**Figure 1 ijms-26-08451-f001:**
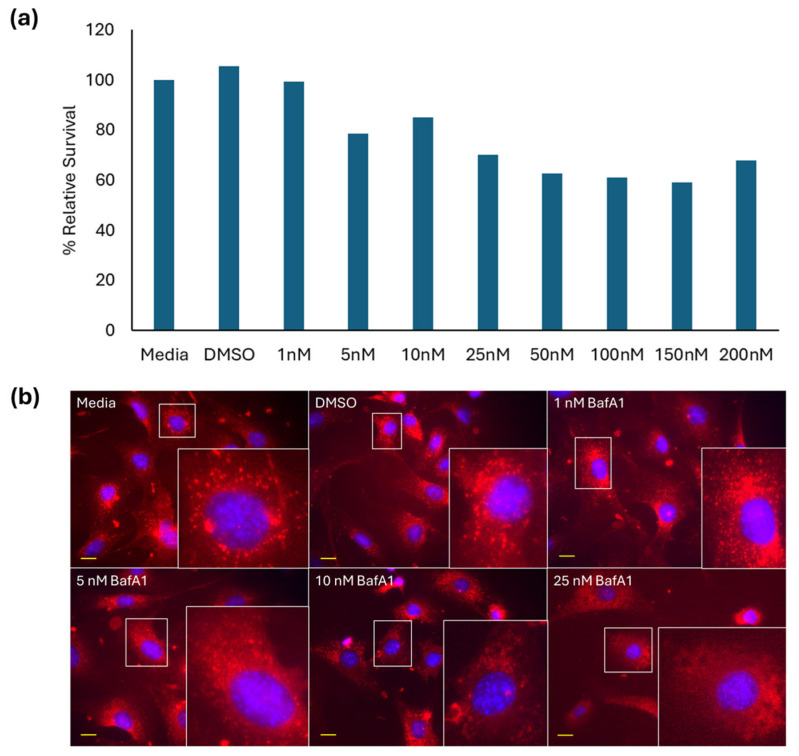
(**a**) BafA1 dose range finding study (n = 1) and (**b**) 40× fluorescent images of FE1 cells after 24 h exposure to BafA1. Lysotracker DND (red) and Hoechst 33342 (blue). Yellow scale bars = 10 µm for each image. Insets represent enlargements of areas in white boxes.

**Figure 2 ijms-26-08451-f002:**
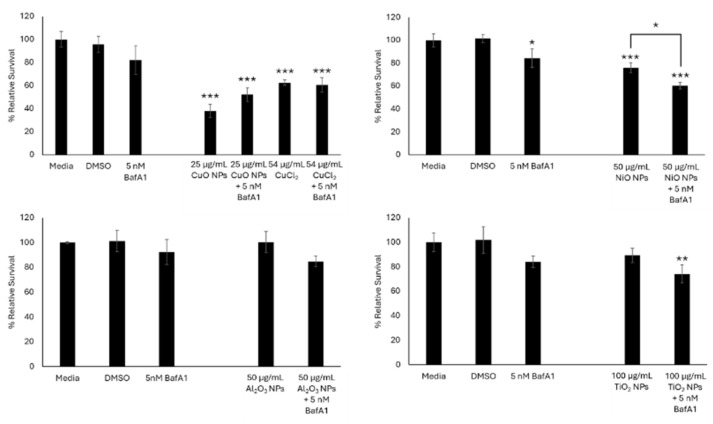
Relative survival of FE1 cells following 24 h treatment with MONPs +/− 5 nM BafA1. Error bars represent standard deviation (n = 3). Significance was determined through a one-way ANOVA with Tukey’s post hoc in the case of a significant result. *: *p* < 0.05. **: *p* < 0.005. ***: *p* < 0.0005.

**Figure 3 ijms-26-08451-f003:**
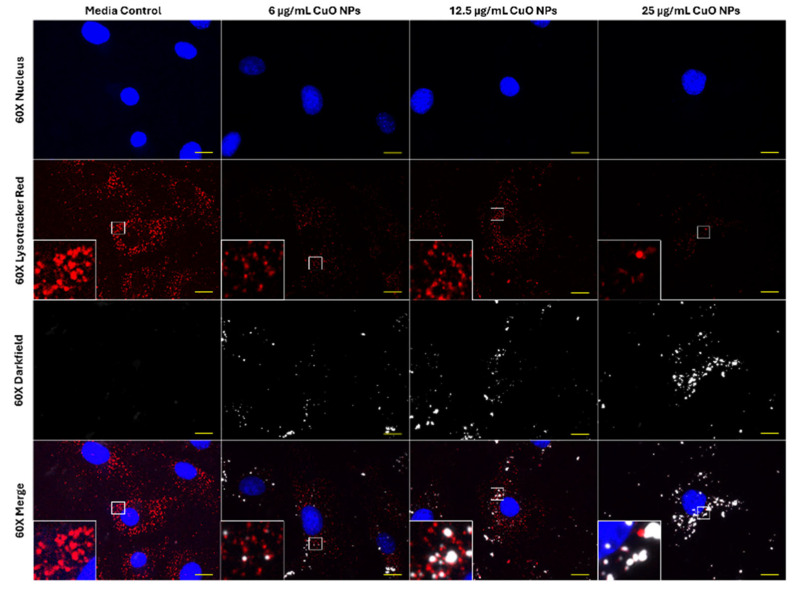
Sixty× enhanced darkfield fluorescent imaging of FE1 cells after 24 h of exposure to blank media or ENMs. Yellow scale bars = 10 µm for each image. Blue: nucleus (Hoechst 33342). Red: acidic vesicles (Lysotracker Red DND-99). White: particles. Insets represent areas highlighted by white boxes.

**Figure 4 ijms-26-08451-f004:**
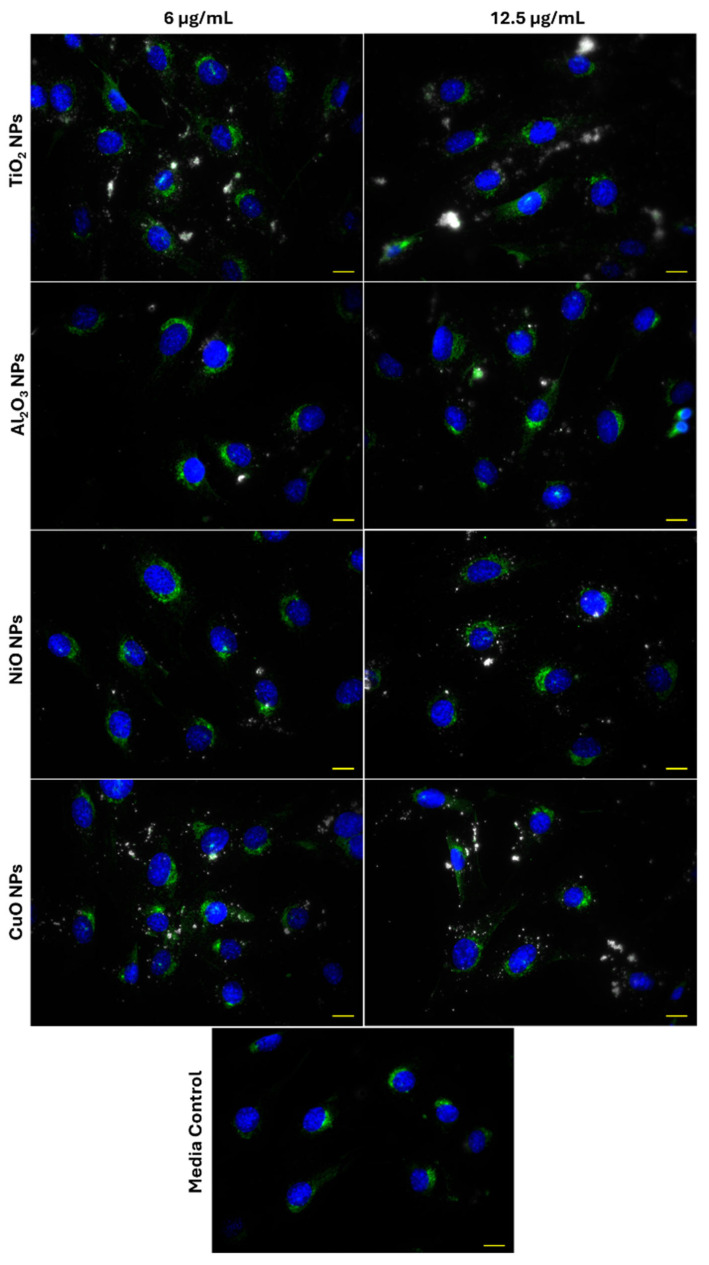
Lysosome distribution in FE1 cells following 24 h exposure to MONPs and blank media. Yellow scale bar = 10 µm in each image. Blue: nucleus (Hoechst 33342). Green: lysosomes (Lamp1). White: MONPs.

**Figure 5 ijms-26-08451-f005:**
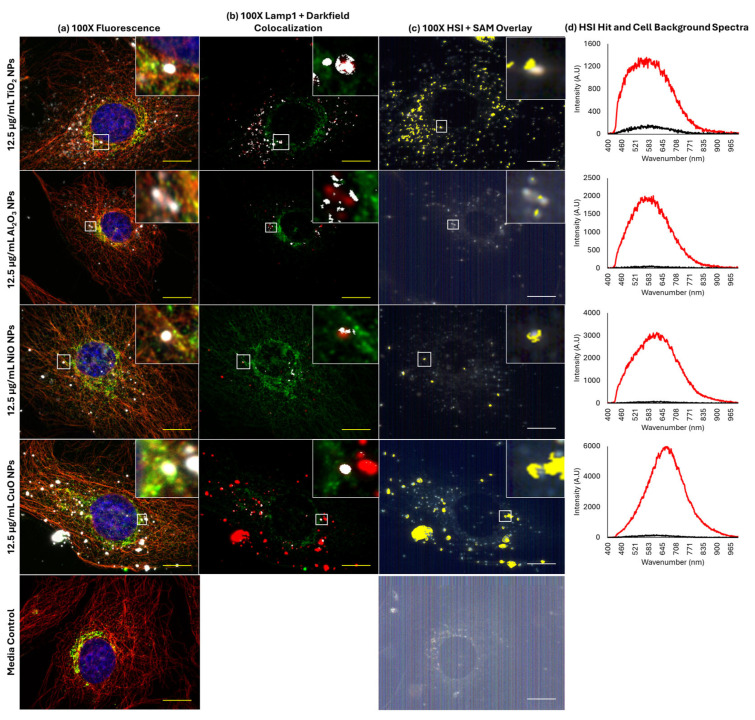
Co-localization of MONPs and lysosomes in FE1 cells following 24 h exposure. Yellow/white scale bars = 10 µm. Insets represent areas highlighted by white boxes. (**a**) Red, α-tubulin; green, lysosomes (Lamp1); blue, nucleus (Hoechst 33342); white, particles. (**b**) Red, particles; green, lysosomes (Lamp1); white, areas of overlap between particles and lysosomes. (**c**) Yellow: areas with spectral match to the MONP of interest. (**d**) Red: particle hit spectrum. Black: cell background spectrum.

**Figure 6 ijms-26-08451-f006:**
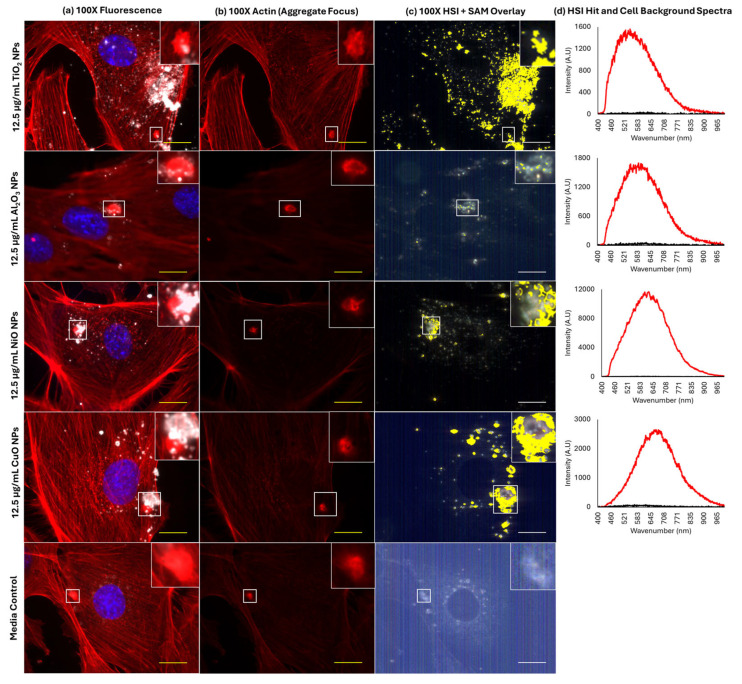
Association of MONPs and actin aggregates in FE1 cells following 24 h exposure. Yellow /white scale bars = 10 µm. Insets represent the enlargement of the area highlighted by white squares. (**a**) Red, F-actin; blue, nucleus (Hoechst 33342); white, particles. The inset represents an enlargement of the area highlighted by white boxes. (**b**) Red: F-actin. (**c**) Yellow: areas with spectral match to the MONP of interest. The inset represents an enlargement of the area highlighted by a white arrow. (**d**) Red: particle hit spectrum. Black: cell background spectrum.

**Figure 7 ijms-26-08451-f007:**
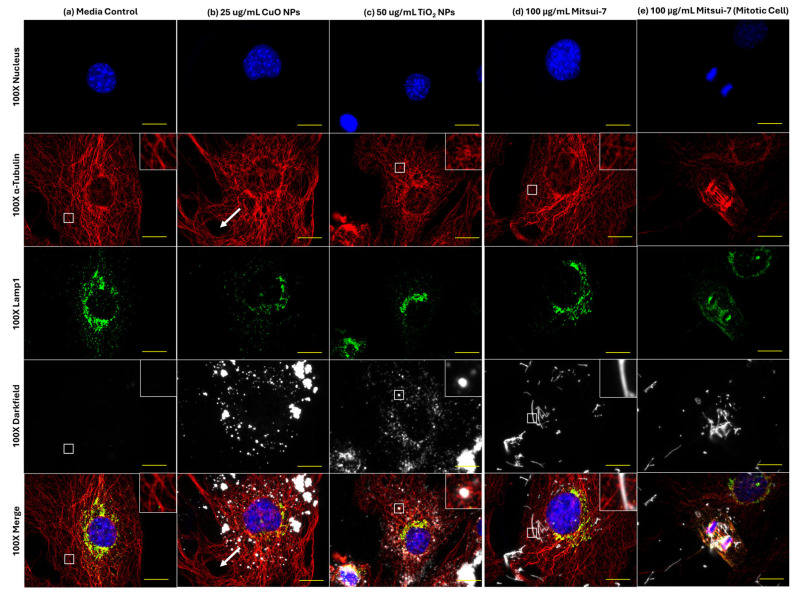
Effects of 24 h of exposure to (**a**) blank media, (**b**) 25 μg/mL CuO NPs, (**c**) 50 μg/mL TiO_2_ NPs, or (**d**,**e**) 100 μg/mL Mitsui-7 MWCNT on lysosomes and α-tubulin structure in FE1 cells. Yellow scale bar = 10 μm in each image. Red: α-tubulin. Green: lysosomes (Lamp1). Blue: nucleus (Hoechst 33342). White: particles or fibers. Insets represent enlarged areas highlighted by the white squares. White arrows highlight a large hole formed in the tubulin structure of a CuO NP-exposed cell.

**Figure 8 ijms-26-08451-f008:**
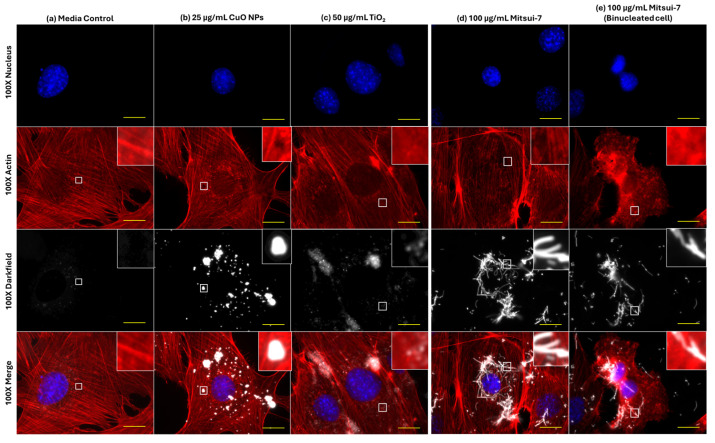
Effects of 24 h of exposure to (**a**) blank media, (**b**) 25 μg/mL CuO NPs, (**c**) 50 μg/mL TiO_2_ NPs, or (**d**,**e**) 100 μg/mL Mitsui-7 MWCNT on the F-actin cytoskeleton in FE1 cells. Yellow scale bars = 10 μm in each image. Red: F-actin. Blue: nucleus. White: particles or fibers. Insets represent an enlargement of the area highlighted by white squares.

**Figure 9 ijms-26-08451-f009:**
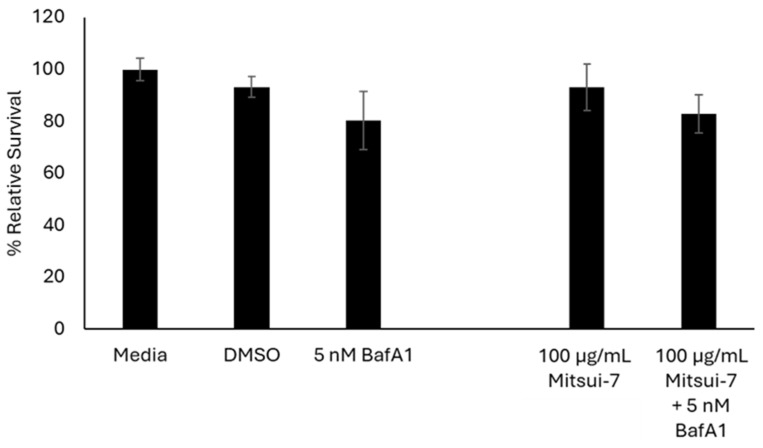
Relative survival of FE1 cells following 24 h treatment with Mitsui-7 +/− 5 nM BafA1. Error bars represent standard deviation (n = 3). Significance was determined through a one-way ANOVA with Tukey’s post hoc in the case of a significant result. No statistically significant differences are seen between the exposures.

**Table 1 ijms-26-08451-t001:** Maximum proportion of darkfield signal co-localizing with Lamp1 signal from three 100× fields of view. Numbers in parentheses indicate the total number of images (out of three for each concentration) where co-localization is seen based on auto-thresholding.

MONP	6 µg/mL	12.5 µg/mL
CuO	0.0183 (1)	0.0691 (1)
NiO	0.3805 (1)	0.044 (1)
Al_2_O_3_	0.0625 (2)	0.0781 (1)
TiO_2_	0.2693 (3)	0.5301 (2)

**Table 2 ijms-26-08451-t002:** Nanomaterials used in the study. Sizes are provided in terms of the length (longest dimension) and width (shortest dimension) of the particles. Numbers in parentheses represent the standard deviation. SSA: specific surface area. D: diameter. L: length. N/A: not available. Dissolution: data pertains to 24 h dissolution in cell culture media at 10 µg/mL from [34,57].

Material	Catalogue Number	Manufacturer	Size (nm)		Aspect Ratio	SSA (m^2^/g)	Dissolution (%)
CuO NPs	544868	Sigma Aldrich (Oakvill, ON, Canada)	64.8 × 45.9 (47 × 28) ^1^		1.39 (0.39) ^1^	10.3 ^5^	11.8
Al_2_O_3_ NPs	544833	Sigma Aldrich (Oakvill, ON, Canada)	23.9 × 10.7 (12 × 7) ^2^		2.63 (1.40) ^2^	145.3 ^5^	1.25
NiO NPs	US3355	US Research Nanomaterials Inc. (Houston, TX, USA)	27.3 × 21.9 (10 × 8) ^2^		1.25 (0.20) ^2^	36.6 ^5^	0.81
TiO_2_ NPs	NIST 1898	National Institute of Standards and Technology (Gaithersburg, MD, USA)	26.8 × 20.9 (9 × 7) ^3^		1.30 (0.26) ^3^	52.7 ^5^	0.17
Mitsui-7	N/A	Mitsui & Co. (Washington, DC, USA)	D: 40–100 ^4^	L: 3000–9400 ^4^	30–235 ^4^	22–28 ^4^	N/A

^1^ Data from [6]. ^2^ Data from [7]. ^3^ Data from [56]. ^4^ Data from [24]. ^5^ Data from [54].

**Table 3 ijms-26-08451-t003:** List of reagents used in this study and associated manufacturers and catalogue numbers.

Reagent	Manufacturer	Catalogue Number
Copper(II) chloride dihydrate	Millipore Sigma (Burlington, ON, Canada)	C3279-100G
Lyophilized Bafilomycin A1	Millipore Sigma (Burlington, ON, Canada)	196000-10UG
100% dimethyl sulfoxide	Fisher Scientific (Ottawa, ON, Canada)	BP231-100
DNAse/RNAse-free ultrapure water	Life Technologies (Burlington, ON, Canada)	10977015
18 mm × 18 mm #1 square glass coverslips	Fisher Scientific (Ottawa, ON, Canada)	12-548A
Formaldehyde solution	Millipore Sigma (Burlington, ON, Canada)	F8775-500ML
Triton X-100	Fisher Scientific (Ottawa, ON, Canada)	BP151-500
Bovine serum albumin	Millipore Sigma (Burlington, ON, Canada)	A9647-100G
Tween-20	Millipore Sigma (Burlington, ON, Canada)	P9416-100ML
ProLong™ Glass Antifade Mountant with NucBlue™ Stain	ThermoFisher (Ottawa, ON, Canada)	P36983
Anti-Lamp1 antibody	Abcam (Waltham, MA, USA)	ab24170
Anti-alpha tubulin antibody	Abcam (Waltham, MA, USA)	ab52866
Phalloidin-iFluor 594	Abcam (Waltham, MA, USA)	ab176757
Goat Anti-Rabbit IgG H&L (Alexa Fluor^®^ 488)	Abcam (Waltham, MA, USA)	ab150077
Goat Anti-Rabbit IgG H&L (Alexa Fluor 568)	Abcam (Waltham, MA, USA)	ab175471
Lysotracker Red DND-99	Thermofisher (Ottawa, ON, Canada)	L7529
Dulbecco’s Modified Eagle’s Medium Nutrient Mixture: F12 HAM (1:1) phenol red-containing medium	Life Technologies (Burlington, ON, Canada)	21041-025
Penicillin-Streptomycin	Life Technologies (Burlington, ON, Canada)	15140122
Fetal bovine serum	Life Technologies (Burlington, ON, Canada)	12483-020
Human epidermal growth factor	Life Technologies (Burlington, ON, Canada)	PHG0311
Dulbecco’s Modified Eagle’s Medium Nutrient Mixture: F12 HAM (1:1) without phenol red	Life Technologies (Burlington, ON, Canada)	21041-025
Trypsin	Life Technologies (Burlington, ON, Canada)	25200-056

**Table 4 ijms-26-08451-t004:** Sonication parameters for ENMs used for exposure. Conc: concentration. Amp: amplitude. AST: active sonication time. DSE: delivered sonication energy. DSE = (P × T)/V, P = power (W), T = time (s), V = volume (mL).

Material	Conc (mg/mL)	Volume (mL)	DSE (J/mL)	% Amp	Frequency	AST (s)
CuO NPs	1	8	109.5	10	8 s on/ 2 s off	144
NiO NPs	5	8	733	60	8 s on/ 2 s off	90
Al_2_O_3_ NPs	5	8	1270	55	8 s on/ 2 s off	180
TiO_2_ NPs	5	50	1013	55	8 s on/ 2 s off	900
Mitsui-7 MWCNTs	1	10.61	550 ^1^/172 ^2^	10	Always on	960 ^1^/300 ^2^

^1^ Initial sonication of stock Mitsui-7. ^2^ Subsequent sonication of thawed primary stock.

## Data Availability

The original contributions presented in this study are included in the article/Appendix A. Further inquiries can be directed to the corresponding author.

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
