# Peer review of "Acute Toxicity of Metal Oxide Nanoparticles—Role of Intracellular Localization In Vitro in Lung Epithelial Cells"

_ijms, 2025, doi:10.3390/ijms26178451_

Round 1
Reviewer 1 Report
Comments and Suggestions for Authors
This study investigated the sequestration of MONPs with varying solubility in lysosomes and the role of lysosomal acidification in their toxicity in FE1 lung epithelial cells, using CuO, NiO, Al2O3, and TiO2 NPs, with Mitsui-7 MWCNTs as contrasts. All MONPs co-localized with lysosomes, with TiO2 NPs showing the greatest co-localization, but only the toxicity of soluble CuO NPs was influenced by lysosomal acidification inhibition. Additionally, MONPs were associated with actin aggregates, with insoluble TiO2 NPs disrupting F-actin and α-tubulin organization, highlighting differential intracellular effects and toxicity mechanisms. Here are some comments for the authors to revise the manuscript.
- For Figure 1a, there is only one test for BafA1 dose-range finding study. The data should be repeated at least 3 times.
- This paper would be more impressive if the authors provided electron microscope images of CuO, NiO, Al2O3, and TiO2 NPs in the supporting information. This data would offer readers a better understanding of the morphologies of the MONPs.
- In the introduction, the sentence “As with all ENM, the toxicity of MONPs is influenced by particle-specific physicochemical properties…” should have more references to reveal the importance.
https://doi.org/10.3390/ijms25031926
Reviewer 2 Report
Comments and Suggestions for Authors
In this study, the authors investigated the potential intracellular sequestering of metal oxide nanoparticles (MONPs) with varying solubility in lysosomes and the role of the acidic lysosomal milieu on toxicity induced by various metal oxide nanoparticles in FE1 lung epithelial cells. This study is quite interesting; however, some comments should be considered to make the manuscript sound as follows:
- Title: It should be modified to include in vitro studies on lung.
- Abstract: key findings and some quantitative data should be highlighted. Line 26: please correct (to fully explore or understand).
- Intro: Line 35: does accidental inhalation represent the main exposure to MONPs?
- Results: Fig. 1a: how many replicates did you apply? Line 105: please correct (an improvement in relative survival or an improvement in survival percentage of…..) and modify figure 2 legend too. Fig. 3: it should be improved. Line 141: please correct (As shown in ….). Fig. 6: it should be improved. Fig. 9: please illustrate statistical analysis (The authors stated statistical analysis in figure legend). Line 242: please correct (Figure 9 shows…..).
- Discussion: Line 273 (From the four not 4). Please do not re-present the results as shown in lines (283-287) since the authors should briefly present the main finding with interpretation and pertinent references or their explanation. Lines 318-325 (this paragraph shows results in details, please summarize for the discussion section). Please do that for the all discussion and remove redundant information. The limitations of this study should be discussed, indicating future perspectives. Pertinent references could be used; for instance, https://doi.org/10.1038/s41598-022-17712-z.
- Methods: Cell culture and other reagents: Please add these reagents in a table in SI. Why did you select FE1 lung cells? Line 530 (18 × 18-1: please revise).
Round 2
Reviewer 2 Report
Comments and Suggestions for Authors
The authors responded to all comments carefully. However, minor comments should be addressed as follows:
- Fig. 9: The author responded to the previous comments as follows:
(The reason no asterisks are present in the figure is because no exposures produced statistically significant differences to the media control).
In this case, please mention in the figure legend that there is no significant difference and illustrate ns in the figure. The authors should also remove *: p <0.05. **: p<0.005. ***: p<0.0005 from the figure legend.
- Line 530: it should be corrected to 18 × 18-1.
